# T-Peak to T-End Interval for Prediction of Positive Response to Ajmaline Challenge Test in Suspected Brugada Syndrome Patients

**DOI:** 10.3390/medsci10040069

**Published:** 2022-12-19

**Authors:** Mananchaya Thapanasuta, Ronpichai Chokesuwattanaskul, Pattranee Leelapatana, Voravut Rungpradubvong, Somchai Prechawat

**Affiliations:** 1Section of Electrophysiology, Division of Cardiovascular Medicine, Department of Medicine, Cardiac Center, Faculty of Medicine, Chulalongkorn University, Bangkok 10330, Thailand; 2Center of Excellence in Arrhythmia Research Chulalongkorn University, Department of Medicine, Faculty of Medicine, Chulalongkorn University, Bangkok 10330, Thailand

**Keywords:** Brugada syndrome, ajmaline, sodium-channel blocker, T-peak to T-end, electrocardiogram

## Abstract

Background: Brugada syndrome (BrS) is diagnosed in patients with ST-segment elevation with coved-type morphology in the right precordial leads, occurring spontaneously or after provocative drugs. Due to electrocardiographic (ECG) inconsistency, provocative drugs, such as sodium-channel blockers, are useful for unmasking BrS. Ajmaline is superior to flecainide and procainamide to provoke BrS. Prolonged T-peak to T-end (TpTe) is associated with an increased risk of ventricular arrhythmia and sudden cardiac death in Brugada syndrome patients. Objective: This study aimed to investigate the predictive value of T-peak to T-end interval and corrected T-peak to T-end interval for predicting the positive response of the ajmaline challenge test in suspected Brugada syndrome patients. Methods: Patients who underwent the ajmaline test in our center were enrolled. Clinical characteristics and electrocardiographic parameters were analyzed, including TpTe, corrected TpTe, QT, corrected QT(QTc) interval, and S-wave duration, compared with the result of the ajmaline challenge test. Results: The study found that TpTe and corrected TpTe interval in suspected BrS patients were not significantly associated with a positive response to the ajmaline challenge test. Conclusions: The T-peak to T-end interval and corrected T-peak to T-end interval could not predict the positive response of the ajmaline challenge test in suspected Brugada syndrome patients.

## 1. Introduction

Brugada syndrome (BrS) is a genetic disorder that affects the cardiac ion channel, mainly the sodium channel. It increases the risk of life-threatening ventricular arrhythmia and sudden cardiac death in patients with a structurally normal heart. The prevalence of Brugada syndrome is 0.36% in Europe and the United States, 1.4% in Japan, and 1.8% in Thailand. The onset of symptoms typically occurs at a mean age of 41 ± 15 years [1,2].

Brugada syndrome is diagnosed in patients with ST-segment elevation with coved-type morphology in the right precordial leads, occurring spontaneously or after provocative drugs. Due to electrocardiographic (ECG) inconsistency, provocative drugs could unmask Brugada syndrome. Ajmaline was superior to flecainide and procainamide to provoke BrS [3]. Ajmaline is an intravenous class I antiarrhythmic drug with a potent sodium channel blocking property. The ajmaline challenge test’s sensitivity and specificity are 80% and 94.4%, respectively [4,5]. In Thailand, the limited use of the ajmaline challenge test has restricted the diagnosis and the assessment of the true prevalence of the disease.

The cellular electrophysiological mechanism of the disease remains to be clarified. Prolonged T-peak to T-end (TpTe) interval is associated with an increased risk of ventricular arrhythmia and sudden cardiac death in Brugada syndrome patients [6]. We hypothesize that T-peak to T-end interval might predict the positivity of the sodium channel challenge test in suspected Brugada syndrome patients.

This study aims to identify whether T-peak to T-end interval and corrected T-peak to T-end interval could predict a positive response in the ajmaline challenge test in suspected Brugada syndrome patients. Furthermore, the electrocardiographic changes following the ajmaline challenge test in diagnosed Brugada syndrome patients, which might contribute to the knowledge of the ambiguous mechanism of Brugada syndrome, were also evaluated.

## 2. Materials and Methods

### 2.1. Study Population and Trial Design

This was a single-center analytical study. The study enrolled all patients aged at least 18 years with a familial screening of Brugada syndrome, unexplained syncope, or resuscitated sudden cardiac arrest (SCA) with suspicious Brugada pattern form. Patients underwent an indicative ajmaline challenge test in King Chulalongkorn Memorial Hospital, Thailand, between August 2018 and December 2021. The study excluded pregnancy, breastfeeding, history of myocardial infarction, allergy to ajmaline, liver disease or SGOT, SGPT above two times the upper limit of normal, concurrent use of tricyclic antidepressant, fluoxetine, lithium, antihistamine, cocaine, propofol, trifluoperazine, measured body temperature above 37.8 or below 36 Celsius, and hypokalemia. Written informed consent was acquired from all patients.

### 2.2. Ethics

The study protocol was approved by the Institutional Review Board of the Faculty of Medicine, Chulalongkorn University, Bangkok, Thailand no. 503/64. No commercial organization was involved in the trial.

### 2.3. Ajmaline Challenge Test Protocol

Twelve-lead electrocardiography using a Philips PageWriter TC70 cardiograph was performed at baseline with leads V1, V2, and V3 placed at the fourth intercostal space (standard position) and the second and third intercostal spaces (Brugada lead position).

During continuous 12-lead ECG monitoring using a Philips PageWriter TC70 cardiograph, all patients were administered intravenous 1 mg/kg ajmaline in 10 min according to the protocol of the Brugada consensus conference. Defibrillator and isoproterenol were prompted for the treatment of ventricular arrhythmia. Intravenous ajmaline infusion was discontinued when Brugada type 1 morphology appeared in at least one right precordial lead, the occurrence of premature ventricular contractions, ventricular arrhythmias, prolongation of QRS duration more than 130 percent of baseline, or presence of higher degree atrioventricular block. Twelve-lead ECGs with leads V1, V2, and V3 in the standard and Brugada lead positions were performed at 1, 5, 10, and 15 min after the ajmaline challenge. After discontinuation of the ajmaline infusion, monitoring was continued for at least 60 min or until normalization of the ST segment. All the patients were observed for 4–6 h after the test.

A positive result included the conversion of type 2, 3 ECG of Brugada pattern to type 1 morphology (coved type) and a J-wave amplitude of more than 2 mm in leads V1, V2, and V3. Clinical characteristics and electrocardiographic parameters were analyzed.

### 2.4. Electrocardiographic Analysis

The T-peak to T-end interval, corrected T-peak to T-end interval (corrected using Bazett’s formula), and S-wave duration were measured in leads II, III, aVF, V2, V5, and V6. The QT interval and corrected QT interval (corrected using Bazett’s formula) were measured. Additional parameters measured were P-wave duration, PR interval, PQ interval, QRS duration, QT peak, T-segment elevation, RR interval, JT interval, terminal activation time (TAD), and ventricular activation time (VAT) in lead II. The values of each parameter were reported as the largest measurement evaluated in each derivation.

The electrocardiography was performed at ten times zoom with an amplitude of 10 mm/mV and a speed of 25 mm/s. Two independent physicians blinded to the clinical status measured ECGs. All measurements were performed using an electronic caliper, Phillips intelliSpace ECG Program. The intra-observer variability using the mean-centered coefficient of variation was 0.3%, and the inter-observer variability using interclass correlation was 0.86.

### 2.5. Statistical Analysis

Baseline clinical and electrocardiographic characteristics were presented as means and standard deviations for continuous variables and counts and percentages for categorical variables. Differences between the two groups were assessed using Fisher’s exact test, independent *t*-test, and Mann–Whitney U test. A *p*-value less than 0.05 was considered statistically significant. Data were analyzed using SPSS version 24.3.

## 3. Results

### 3.1. Baseline Characteristics

Between August 2018 and December 2021, a total of 16 consecutive patients underwent an indicative ajmaline challenge test in King Chulalongkorn Memorial Hospital. The mean age was 40 ± 15 years. Fifteen patients (93.7%) were men. One patient (6.3%) received an implantable cardioverter-defibrillator. An SCN5A mutation was identified in one patient (6.3%). The indications for the ajmaline challenge test were familial screening 37.5%, unexplained syncope 43.8%, and resuscitated SCA 18.8%. The mean T-peak to T-end interval was II 113.5 ± 26.8 ms, III 91.0 ± 33.1 ms, avF 105.0 ± 22.5 ms, V2 115.9 ± 21.9 ms, V5 108.4 ± 22.1, and V6 108.9 ± 19.5 ms. The mean corrected T-peak to T-end interval was II 118.8 ± 33.7 ms, III 95.6 ± 39.7 ms, avF 110.4 ± 31.6 ms, V2 120.9 ± 27.0 ms, V5 113.6 ± 30.2, and V6 114.3 ± 28.5 ms. The mean QT interval was 404.4 ± 29.1 ms. The mean corrected QT interval was 415.7 ± 29.6 ms. The mean S-wave duration was II 41.2 ± 14.2 ms, III 37.8 ± 19.5 ms, avF 35.5 ± 19.1 ms, V2 52.5 ± 19.7 ms, V5 41.9 ± 21.1 ms, and V6 37.4 ± 21.4 ms. The patient characteristics are summarized in Table 1.

### 3.2. Electrocardiographic Parameters According to Ajmaline Challenge Test Result

The ajmaline challenge test showed a positive response in 12 out of 16 patients (75%). We found that the TpTe and corrected TpTe intervals in suspected BrS patients were not significantly associated with a positive response in the ajmaline challenge test (TpTe: II Neg 96.2 ± 8.7 ms vs. Pos 119.2 ± 28.6 ms, *p* = 0.133, III Neg 88.7 ± 22.7 ms vs. Pos 91.7 ± 36.8 ms, *p* = 0.882, avF Neg 92.5 ± 18.8 ms vs. Pos 109.2 ± 22.8 ms, *p* = 0.211, V2 Neg 127.2 ± 29.0 ms vs. Pos 112.1 ± 19.0 ms, *p* = 0.243, V5 Neg 102.2 ± 29.5 ms vs. Pos 110.5 ± 20.3 ms, *p* = 0.538, V6 Neg 114.2 ± 30.7 ms vs. Pos 107.2 ± 15.8 ms, *p* = 0.549, cTpTe: II Neg 102.5 ± 16.0 ms vs. Pos 124.2 ± 36.7 ms, *p* = 0.379, III Neg 93.4 ± 21.5 ms vs. Pos 96.3 ± 44.9 ms, 0.906, avF Neg 98.8 ± 24.8 ms vs. Pos 114.3 ± 33.6 ms, *p* = 0.415, V2 Neg 135.0 ± 34.0 ms vs. Pos 116.2 ± 24.0 ms, *p* = 0.239, V5 Neg 108.3 ± 32.9 ms vs. Pos 115.3 ± 30.6 ms, *p* = 0.701, V6 Neg 122.7 ± 40.4 ms vs. Pos 111.5 ± 25.1 ms, *p* = 0.515) (Figure 1).

The QT interval, the corrected QT interval, and the S-wave duration in suspected BrS patients were also not significantly associated with a positive response to the test (QT: Neg 412.2 ± 35.0 ms vs. Pos 401.8 ± 28.2 ms, *p* = 0.554, QTc: Neg 436.2 ± 33.0 ms vs. Pos 408.8 ± 26.3 ms, *p* = 0.110, S: II Neg 41.5 ± 16.8 ms vs. Pos 42.0 ± 14.1 ms, *p* = 0.954, III Neg 33.7 ± 11.1 ms vs. Pos 39.2 ± 21.9 ms, *p* = 0.953, avF Neg 26.5 ± 8.7 ms vs. Pos 38.5 ± 20.9 ms, *p* = 0.521, V2 Neg 43.2 ± 13.5 ms vs. Pos 55.6 ± 0.9 ms, *p* = 0.293, V5 Neg 31.0 ± 8.0 ms vs. Pos 45.5 ± 23.1 ms, *p* = 0.446, V6 Neg 24.0 ± 3.3 ms vs. Pos 41.9 ± 23.2 ms, *p* = 0.262). According to the ajmaline challenge test results, the electrocardiographic parameters are summarized in Table 2.

### 3.3. Electrocardiographic Parameters at Baseline and End of the Test

After the ajmaline challenge test, the corrected QT interval and S-wave duration (lead II, V5) were significantly increased (QTc: Baseline 415.7 ± 29.6 ms vs. End of test 465.9 ± 34.3 ms, Mean difference 52.5 ± 38.2 ms, *p* < 0.001, S: II Baseline 41.2 ± 14.2 ms vs. End of test 57.8 ± 33.6 ms, Mean difference 17.5 ± 29.3 ms, *p* = 0.037, V5 Baseline 41.9 ± 21.1 ms vs. End of test 10.5 ± 21.1 ms, Mean difference 10.5 ± 21.1 ms, *p* = 0.029). The electrocardiographic parameters after the ajmaline challenge test are summarized in Table 3.

Additionally, P-wave duration, PR interval, PQ interval, QRS duration, and ventricular activation time were significantly increased after the ajmaline testing (P: Baseline 110.4 ± 18.4 ms vs. End of test 139.9 ± 27.2 ms, Mean difference 28.3 ± 32.7 ms, *p* = 0.005, PR: Baseline 174.9 ± 27.8 ms vs. End of test 201.7 ± 24.1 ms, Mean difference 25.7 ± 30.0 ms, *p* = 0.005, PQ: Baseline 171.4 ± 24.7 ms vs. End of test 218.4 ± 29.9 ms, Mean difference 44.5 ± 37.8 ms, *p* < 0.001, QRSd: Baseline 103.4 ± 14.9 ms vs. End of test 128.9 ± 24.0 ms, Mean difference 25.8 ± 27.1 ms, *p* = 0.002, VAT: Baseline 33.9 ± 5.2 ms vs. End of test 41.2 ± 6.4 ms, Mean difference 6.7 ± 6.9 ms, *p* = 0.005). The T-wave height was significantly reduced after the test (Baseline 0.4 ± 0.1 mV vs. End of test 0.3 ± 0.1 mV, Mean difference −0.06 ± 0.1 mV, *p* = 0.045).

### 3.4. Electrocardiographic Changes following the Ajmaline Challenge Test in Diagnosed Brugada Syndrome Patients

In a total of 12 Brugada syndrome patients diagnosed with a positive ajmaline challenge test, corrected QT interval and S-wave duration in lead II were significantly increased following the ajmaline challenge test (QTc: Baseline 408.8 ± 26.3 ms vs. End of test 465.1 ± 38.2 ms, mean difference 56.2 ± 39.8 ms, *p* = 0.006, S: II Baseline 42.0 ± 14.1 ms vs. End of test 64.9 ± 34.0 ms, mean difference 22.9 ± 30.5 ms, *p* = 0.025) as shown in Table 4 and Figure 2.

In addition, P-wave duration, PR interval, PQ interval, QRS duration, and ventricular activation time were significantly increased in diagnosed Brugada syndrome patients after the ajmaline testing (P: Baseline 114.2 ± 17.3 ms vs. End of test 139.8 ± 29.8 ms, Mean difference 25.7 ± 35.4 ms, *p* = 0.029, PR: Baseline 180.8 ± 23.2 ms vs. End of test 205.2 ± 24.4 ms, Mean difference 24.3 ± 31.7 ms, *p* = 0.022, PQ: Baseline 177.0 ± 20.8 ms vs. End of test 216.8 ± 32.9 ms, Mean difference 39.8 ± 35.9 ms, *p* = 0.003, QRSd: Baseline 107.1 ± 12.6 ms vs. End of test 131.5 ± 25.8 ms, Mean difference 24.4 ± 30.3 ms, *p* = 0.018, VAT: Baseline 34.2 ± 5.1 ms vs. End of test 42.2 ± 6.6 ms, Mean difference 8.0 ± 7.0 ms, *p* = 0.002).

### 3.5. Complications

Among the 16 performed ajmaline challenge tests, no serious adverse event was observed in the trial.

## 4. Discussion

Our study is the first analytical study to demonstrate the practice of the ajmaline challenge test in suspected Brugada syndrome patients in Thailand. A prolonged T-peak to T-end interval was related to an increased risk of ventricular arrhythmia and sudden cardiac death, including appropriate ICD therapy [6,7,8]. We hypothesized that T-peak to T-end interval could predict a positive response in the ajmaline challenge test in patients with suspected Brugada syndrome. However, the main finding in our trial is that T-peak to T-end interval, corrected T-peak to T-end interval (including QT interval and corrected QT interval), and S-wave duration were not significantly related to positivity in the ajmaline challenge test. Consequently, the ajmaline challenge test remains the definite diagnostic test in clinical suspicion of the disease without reliable EKG markers for positive response.

Our study demonstrated the safety of the ajmaline challenge test. There was no reported complication in this study, in which the incidence was lower than that of previous trials, reporting the development of symptomatic ventricular arrhythmias at 0.15–1.60 percent [9,10].

Resulting from the small population in this trial, this should be interpreted with caution. Therefore, we encourage the diagnostic use of the intravenous ajmaline challenge test under continuous ECG monitoring with prompted defibrillator and isoproterenol in clinically suspected patients in Thailand.

We also observed a significant change in many electrocardiographic parameters after ajmaline infusion: P-wave duration, PR interval, PQ interval, QRS duration, ventricular activation time, S-wave duration (leads II, V5), and corrected QT interval. The electrocardiographic change implies that ajmaline affects both depolarization and repolarization of the cardiac cycle. This result contradicts the concept that ajmaline affects only the sodium channels and notes the additional influence of ajmaline on potassium and calcium channels. A previous study investigated that ajmaline had multiple mechanisms of action, including an inhibitory effect on sodium channels (INa), L-type calcium channels (ICa-L), and transient outward potassium channels (Ito) in rat right-ventricular myocytes [11,12]. Ajmaline also inhibits calcium channels (ICa) and inwardly rectifying potassium channels (IK1) in guinea pig ventricular cardiomyocytes [11,12].

The pathophysiological mechanism behind Brugada syndrome is still debated. There are two main electrophysiological theories of the disease. The repolarization theory explains the transmural dispersion of repolarization between the endocardium and the epicardium of the right ventricle [13]. On the contrary, the depolarization theory focuses on structural abnormalities and delayed conduction of the right ventricle [14]. We detected an essential increase in the corrected QT interval, representing the repolarization of the cardiac cycle, and S-wave duration (lead II), representing the depolarization, following the ajmaline challenge test in diagnosed Brugada syndrome patients. The result might imply that Brugada syndrome influences both depolarization and repolarization.

### Study Limitations

Even though this pilot study provided novel data about the ajmaline challenge test in Thailand, it remains a relatively small, single-center, nonrandomized analytical study. Hence, it may not be possible to extrapolate the results of our study to the broader population. Furthermore, the small size of the enrolled sample population limits statistical power to demonstrate the predictivity of T-peak to T-end interval and corrected T-peak to T-end interval. However, the absence of difference might be related to the minor differences in overall electrocardiographic parameters.

## 5. Conclusions

This study found that the T-peak to T-end interval and corrected T-peak to T-end interval could not predict positive response to the ajmaline challenge test in suspected Brugada syndrome patients. The EKG changes after ajmaline testing support the influence of both depolarization and repolarization on the Brugada syndrome patients.

## Figures and Tables

**Figure 1 medsci-10-00069-f001:**
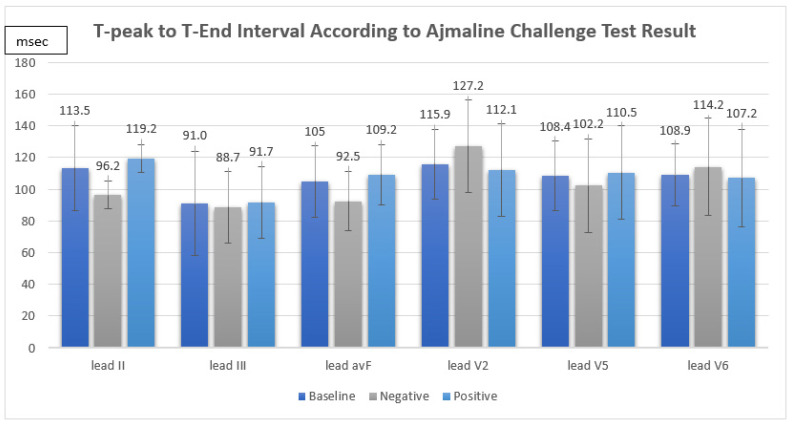
T-peak to T-end interval according to ajmaline challenge test result.

**Figure 2 medsci-10-00069-f002:**
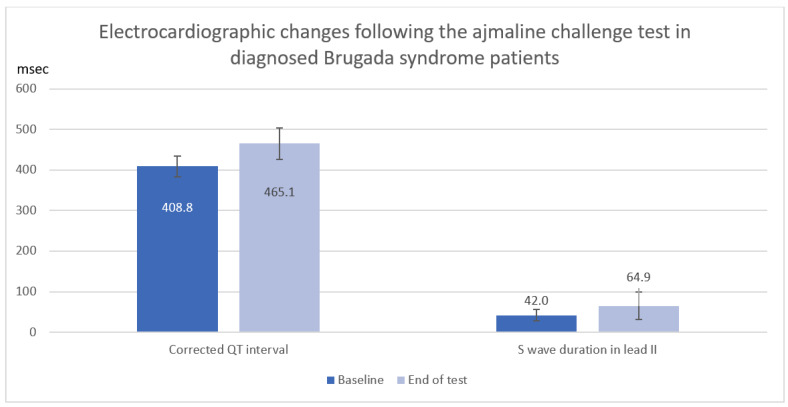
Electrocardiographic changes (Corrected QT interval and S wave duration) after ajmaline infusion in diagnosed Brugada syndrome patients.

**Table 1 medsci-10-00069-t001:** Baseline Characteristics.

Baseline Characteristics	*n* (%) or Mean ± SD	Ajmaline Challenge Test		*p*-Value
		Negative	Positive	
		(*n* = 4)	(*n* = 12)	
Age, years	40 ± 15	34 ± 18	42 ± 14	0.375
Male	15 (93.7%)	3 (75.0%)	12 (100.0%)	0.250
Indication				
Familial screening of BrS	6 (37.5%)	2 (50.0%)	8 (66.7%)	0.604
Unexplained syncope	7 (43.8%)	2 (50.0%)	7 (58.3%)	1.000
Resuscitated sudden cardiac arrest	3 (18.8%)	4 (100.0%)	9 (75.0%)	0.529
Treatment				
Betablocker	1 (6.3%)	3 (75.0%)	12 (100.0%)	0.250
AICD	1 (6.3%)	0 (0.0%)	1 (8.3%)	1.000
SCN5A mutation	1 (6.3%)	0 (0.0%)	1 (8.3%)	1.000
T-peak to T-end interval, ms				
II	113.5 ± 26.8	96.2 ± 8.7	119.2 ± 28.6	0.133
III	91.0 ± 33.1	88.7 ± 22.7	91.7 ± 36.8	0.882
avF	105.0 ± 22.5	92.5 ± 18.8	109.2 ± 22.8	0.211
V2	115.9 ± 21.9	127.2 ± 29.0	112.1 ± 19.0	0.243
V5	108.4 ± 22.1	102.2 ± 29.5	110.5 ± 20.3	0.538
V6	108.9 ± 19.5	114.2 ± 30.7	107.2 ± 15.8	0.549
Corrected T-peak to T-end interval, ms	118.8 ± 33.7			
II	95.6 ± 39.7	102.5 ± 16.0	124.2 ± 36.7	0.379
III	110.4 ± 31.6	93.4 ± 21.5	96.3 ± 44.9	0.906
avF	120.9 ± 27.0	98.8 ± 24.8	114.3 ± 33.6	0.415
V2	113.6 ± 30.2	135.0 ± 34.0	116.2 ± 24.0	0.239
V5	114.3 ± 28.5	108.3 ± 32.9	115.3 ± 30.6	0.701
V6	113.5 ± 26.8	122.7 ± 40.4	111.5 ± 25.1	0.515
QT interval, ms Corrected QT interval, ms	404.4 ± 29.1	412.2 ± 35.0	401.8 ± 28.2	0.554
	415.7 ± 29.6	436.2 ± 33.0	408.8 ± 26.3	0.110
S-wave duration, ms				
II	41.2 ± 14.2	41.5 ± 16.8	42.0 ± 14.1	0.954
III	37.8 ± 19.5	33.7 ± 11.1	39.2 ± 21.9	0.953
avF	35.5 ± 19.1	26.5 ± 8.7	38.5 ± 20.9	0.521
V2	52.5 ± 19.7	43.2 ± 13.5	55.6 ± 20.9	0.293
V5	41.9 ± 21.1	31.0 ± 8.0	45.5 ± 23.1	0.446
V6	37.4 ± 21.4	24.0 ± 3.3	41.9 ± 23.2	0.262

AICD = Automated implantable cardioverter defibrillator.

**Table 2 medsci-10-00069-t002:** Electrocardiographic parameters according to ajmaline challenge test results.

Variables	Negative Test	Positive Test	*p*-Value
	(*n* = 4)	(*n* = 12)	
T-peak to T-end interval, ms			
II	96.2 ± 8.7	119.2 ± 28.6	0.133
III	88.7 ± 22.7	91.7 ± 36.8	0.882
avF	92.5 ± 18.8	109.2 ± 22.8	0.211
V2	127.2 ± 29.0	112.1 ± 19.0	0.243
V5	102.2 ± 29.5	110.5 ± 20.3	0.538
V6	114.2 ± 30.7	107.2 ± 15.8	0.549
Corrected T-peak to T-end interval, ms			
II	102.5 ± 16.0	124.2 ± 36.7	0.379
III	93.4 ± 21.5	96.3 ± 44.9	0.906
avF	98.8 ± 24.8	114.3 ± 33.6	0.415
V2	135.0 ± 34.0	116.2 ± 24.0	0.239
V5	108.3 ± 32.9	115.3 ± 30.6	0.701
V6	122.7 ± 40.4	111.5 ± 25.1	0.515
QT interval, ms	412.2 ± 35.0	401.8 ± 28.2	0.554
Corrected QT interval, ms	436.2 ± 33.0	408.8 ± 26.3	0.110
S-wave duration, ms			
II	41.5 ± 16.8	42.0 ± 14.1	0.954
III	33.7 ± 11.1	39.2 ± 21.9	0.953
avF	26.5 ± 8.7	38.5 ± 20.9	0.521
V2	43.2 ± 13.5	55.6 ± 20.9	0.293
V5	31.0 ± 8.0	45.5 ± 23.1	0.446
V6	24.0 ± 3.3	41.9 ± 23.2	0.262

Values are presented as mean ± SD.

**Table 3 medsci-10-00069-t003:** Electrocardiographic parameters at baseline and end of the test.

Variables	Baseline	End of the Test	Difference	*p*-Value
T-peak to T-end interval, ms				
II	113.5 ± 26.8	114.9 ± 22.2	0.13 ± 27.5	1.000
III	91.0 ± 33.1	88.7 ± 35.7	−4.5 ± 27.4	0.532
avF	105.0 ± 22.5	107.5 ± 28.9	3.2 ± 33.1	0.714
V2	115.9 ± 21.9	124.5 ± 61.3	6.7 ± 62.6	0.687
V5	108.4 ± 22.1	115.1 ± 19.1	3.5 ± 25.9	0.612
V6	108.9 ± 19.5	104.1 ± 14.1	−1.7 ± 23.7	0.790
Corrected T-peak to T-end interval, ms				
II	118.8 ± 33.7	124.9 ± 36.1	5.2 ± 41.2	0.561
III	95.6 ± 39.7	96.3 ± 45.4	−1.3 ± 33.2	0.876
avF	110.4 ± 31.6	114.3 ± 39.3	5.1 ± 45.1	0.666
V2	120.9 ± 27.0	136.3 ± 72.5	13.9 ± 72.4	0.470
V5	113.6 ± 30.2	124.8 ± 35.2	8.2 ± 38.8	0.428
V6	114.3 ± 28.5	113.4 ± 33.3	3.1 ± 34.7	0.731
QT interval, ms Corrected QT interval, ms	404.4 ± 29.1	415.2 ± 29.9	10.6 ± 27.6	0.160
	415.7 ± 29.6	465.9 ± 34.3	52.5 ± 38.2	<0.001
S-wave duration, ms				
II	41.2 ± 14.2	57.8 ± 33.6	17.5 ± 29.3	0.037
III	37.8 ± 19.5	58.1 ± 52.0	20.1 ± 50.7	0.108
avF	35.5 ± 19.1	43.1 ± 20.3	7.9 ± 14.8	0.060
V2	52.5 ± 19.7	46.1 ± 15.0	−5.8 ± 24.5	0.375
V5	41.9 ± 21.1	52.8 ± 19.6	10.5 ± 21.1	0.029
V6	37.4 ± 21.4	45.8 ± 19.6	7.5 ± 18.6	0.137

Values are presented as mean ± SD.

**Table 4 medsci-10-00069-t004:** Electrocardiographic changes after ajmaline infusion in diagnosed Brugada syndrome patients.

Variables	Baseline	End of the Test	Difference	*p*-Value
T-peak to T-end interval, ms				
II	119.2 ± 28.6	113.5 ± 24.0	−5.7 ± 26.2	0.423
III	91.7 ± 36.8	92.2 ± 37.4	0.5 ± 26.5	0.875
avF	109.2 ± 22.8	107.5 ± 28.1	−1.7 ± 33.6	0.657
V2	112.1 ± 19.0	131.5 ± 64.3	19.4 ± 62.6	0.432
V5	110.5 ± 20.3	116.1 ± 16.7	5.6 ± 27.6	0.456
V6	107.2 ± 15.8	105.0 ± 14.6	−2.2 ± 25.1	0.844
Corrected T-peak to T-end interval, ms				
II	124.2 ± 36.7	121.7 ± 38.9	−2.5 ± 41.0	0.875
III	96.3 ± 44.9	98.8 ± 48.3	2.6 ± 34.5	1.000
avF	114.3 ± 33.6	112.1 ± 39.0	−2.2 ± 45.2	0.638
V2	116.2 ± 24.0	142.5 ± 76.7	26.3 ± 75.0	0.272
V5	115.3 ± 30.6	124.0 ± 35.2	8.7 ± 43.0	0.480
V6	111.5 ± 25.1	112.8 ± 32.7	1.3 ± 37.2	0.875
QT interval, ms Corrected QT interval, ms	401.8 ± 28.2	415.8±32.8	14.0 ± 27.5	0.126
	408.8 ± 26.3	465.1 ± 38.2	56.2 ± 39.8	0.006
S-wave duration, ms				
II	42.0 ± 14.1	64.9 ± 34.0	22.9 ± 30.5	0.025
III	39.2 ± 21.9	65.8 ± 55.7	26.7 ± 55.1	0.056
avF	38.5 ± 20.9	46.6 ± 20.9	8.1 ± 16.0	0.102
V2	55.6 ± 20.9	48.1 ± 16.2	−7.5 ± 27.3	0.362
V5	45.5 ± 23.1	56.9 ± 19.6	11.4 ± 23.7	0.071
V6	41.9 ± 23.2	48.7 ± 20.1	6.8 ± 20.3	0.241

Values are presented as mean ± SD.

## Data Availability

Not applicable.

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
