# Peer review of "T-Peak to T-End Interval for Prediction of Positive Response to Ajmaline Challenge Test in Suspected Brugada Syndrome Patients"

_medsci, 2022, doi:10.3390/medsci10040069_

Round 1

Reviewer 1 Report

1. - The authors should ensure the keywords are previously mentioned in the abstract section. "SODIUM-CHANNEL BLOCKER" is not in the abstract. 

2. - Graphics need to have standard deviation errors bars

3. - Conclusion need to be more complete and should refer the main points highlighted in the discussion section. 

4. - The authors should revise the reference section: References need to be more up to date and in large number. 

Author Response

Reviewer 1

Comments and Suggestions for Authors

  1. - The authors should ensure the keywords are previously mentioned in the abstract section. "SODIUM-CHANNEL BLOCKER" is not in the abstract.

Response: We appreciate this observation. The term “sodium-channel blocker” has been inserted into the abstract accordingly.

Due to electrocardiographic (ECG) inconsistency, provocative drugs, sodium-channel blockers, are useful for unmasking BrS.”

  1. - Graphics need to have standard deviation errors bars

Response: Updated figures (figure 1 and 2) have been inserted into the revised manuscript.

  1. - Conclusion need to be more complete and should refer the main points highlighted in the discussion section.

Response: Thank you for this comment. We have inserted out secondary research question findings into the conclusion section as follows:

“The study found that the T-peak to T-end interval and corrected T-peak to T-end interval could not predict the positive response of the ajmaline challenge test in suspected Brugada syndrome patients. The EKG changes after Ajmaline test support the influence of both depolarization and repolarization on the Brugada syndrome patients.”

  1. - The authors should revise the reference section: References need to be more up to date and in large number.

Response: Thank you for this comment. We have been working on relevant literature and adding more citation in the revised manuscript.

Reviewer 2 Report

Thapanasuta et al evaluated in a single centre study the predictive value of T-peak to T-end interval and corrected T-peak to T-end interval for predicting the positive response of the ajmaline challenge test in suspected Brugada syndrome patients. Authors found that T-peak to T-end interval and corrected T-peak to T-end interval could not predict the positive response of the ajmaline challenge test in suspected Brugada syndrome patients. I found this study interesting but some major issues should be addressed.

·        It would be necessary to increase the sample number in order to state that there was no difference between the two groups. 16 patients is a very small number to make any conclusions.

·        Instead of analysing the values of each individual lead, it would be interesting to compare the highest value of all leads.

·        It would be necessary to report in the methods how the values of each individual ECG parameter were reported: was the largest measurement taken in each derivation, or the average? And was the average reported over how many samples?

·        As the analyses were performed by two physicians, it would be necessary to perform an analysis of inter- and intra-observer variability

Author Response

Reviewer 2

Thapanasuta et al evaluated in a single centre study the predictive value of T-peak to T-end interval and corrected T-peak to T-end interval for predicting the positive response of the ajmaline challenge test in suspected Brugada syndrome patients. Authors found that T-peak to T-end interval and corrected T-peak to T-end interval could not predict the positive response of the ajmaline challenge test in suspected Brugada syndrome patients. I found this study interesting but some major issues should be addressed.

  • It would be necessary to increase the sample number in order to state that there was no difference between the two groups. 16 patients is a very small number to make any conclusions.

Response: Thank you for this comment.  Whereas our study was a small sample, we actually need more enrolled subjects to draw an affirm conclusion in the further study. Patients in this study have a high pretest probability of Brugada syndrome. These enrollment criteria make our cohort unique in demographic data. The limitation of this kind of study, a very selective population, is the small number of enrolled patients, as the reviewer commented. Similarly, the prior studies conducted to explore the mechanism of Brugada syndrome have a similar number of subjects, ranging from 14-35 patients. [1-3] To further analyze the statistical power, we also did an additional power analysis (using n =4, d=3, sig. level = 0.1) which demonstrated a power at 0.98, a probability of avoiding type II errors. This is an acceptable power for the given circumstance.

"The absence of evidence does not mean evidence of absence" is still a status quo that applies to our study. We emphasize this limitation in the discussion section as follows:

"Hence, it may not be possible to extrapolate the results of our study to the broader population."

References

  1. Wolpert C, Echternach C, Veltmann C, Antzelevitch C, Thomas GP, Spehl S, et al. Intravenous drug challenge using flecainide and ajmaline in patients with Brugada syndrome. Heart Rhythm. 2005;2(3):254-60.
  2. Prasad S, Namboodiri N, Thajudheen A, Singh G, Prabhu MA, Abhilash SP, et al. Flecainide challenge test: Predictors of unmasking of type 1 Brugada ECG pattern among those with non-type 1 Brugada ECG pattern. Indian Pacing Electrophysiol J. 2016;16(2):53-8.
  3. Dubner S, Azocar D, Gallino S, Cerantonio AR, Muryan S, Medrano J, et al. Single oral flecainide dose to unmask type 1 Brugada syndrome electrocardiographic pattern. Ann Noninvasive Electrocardiol. 2013;18(3):256-61.

  • Instead of analysing the values of each individual lead, it would be interesting to compare the highest value of all leads.

Response: Thank you for the comment. We did not intend to confuse the readers. We added the following statement into method section for a clarification.

The values of each parameter were reported as the largest measurement evaluated in each derivation.”

  • It would be necessary to report in the methods how the values of each individual ECG parameter were reported: was the largest measurement taken in each derivation, or the average? And was the average reported over how many samples?

Response: Thank you for this comment. We have added the following sentence for further clarification on method section.

The values of each parameter were reported as the largest measurement evaluated in each derivation.”

  • As the analyses were performed by two physicians, it would be necessary to perform an analysis of inter- and intra-observer variability

Response: Thank you for pinpointing this important information we missed in the original version. We have added inter- and intra-observer variation in the method section.

The Intra-observer variability using the mean-centered coefficient of variation was 0.3%, and the inter-observer variability using interclass correlation was 0.86.”

Round 2

Reviewer 2 Report

the authors responded adequately to my comments.  I have no further comments to address.